# Vertical Scanning Interferometry for Label-Free Detection of Peptide-Antibody Interactions

**DOI:** 10.3390/ht8020007

**Published:** 2019-03-27

**Authors:** Andrea Palermo, Richard Thelen, Laura K. Weber, Tobias Foertsch, Simone Rentschler, Verena Hackert, Julia Syurik, Alexander Nesterov-Mueller

**Affiliations:** Institute of Microstructure Technology, Karlsruhe Institute of Technology (KIT), 76131 Baden-Württemberg, Germany

**Keywords:** peptide array, vertical scanning interferometry, atomic force microscopy, serum antibodies

## Abstract

Peptide microarrays are a fast-developing field enabling the mapping of linear epitopes in the immune response to vaccinations or diseases and high throughput studying of protein-protein interactions. In this respect, a rapid label-free measurement of protein layer topographies in the array format is of great interest but is also a great challenge due to the extremely low aspect ratios of the peptide spots. We have demonstrated the potential of vertical scanning interferometry (VSI) for a detailed morphological analysis of peptide arrays and binding antibodies. The VSI technique is shown to scan an array area of 5.1 square millimeters within 3–4 min at a resolution of 1.4 μm lateral and 0.1 nm vertical in the full automation mode. Topographies obtained by VSI do match the one obtained by AFM measurements, demonstrating the accuracy of the technique. A detailed topology of peptide-antibody layers on single spots was measured. Two different measurement regions are distinguished according to the antibody concentration. In the case of weakly diluted serum, the thickness of the antibody layer is independent of the serum dilution and corresponds to the physical thickness of the accumulated antibody layer. In strongly diluted serum, the thickness measured via VSI is linearly proportional to the serum dilution.

## 1. Introduction

The study of peptide-protein interactions in high-density array formats is an efficient way to investigate the binding specificity of protein interaction domains such as WW, SH3, and PDZ domains [1], and to develop novel diagnostics based, for instance, on the detection of specific serum antibodies [2]. The array format enables multiplexed high-throughput assays and requires minimal sample volume [3]. Hereby, the binding events between the proteins and the peptide spots assembled on a substrate are frequently detected using fluorescent labels. Although the labeling of proteins is a well-established procedure, important information about the interactions between biological molecules and the homogeneity of the binding events can be lost. In addition, the labels can influence the protein affinity that may lead to unspecific binding. However, direct application of the known label-free detection methods, such as surface plasmon resonance methods [4,5,6], reflectometric interference spectroscopy [7], or resonator-like microstructures [8], is not possible for the surfaces on which commercially available high-density peptide arrays are synthesized. The transfer of the peptide spots to the standard free-label detection surfaces from a synthesis surface seems to be a very complicated procedure due to the spreading of the peptides cleaved and their amino-acid-dependent diffusion and adsorption [9]. 

Parallel to developments of high-density arrays, optical scanning techniques such as vertical scanning interferometry (VSI) have experienced huge progress in the last decade in terms of scanning velocity and scanning area. Imaging of protein layers with an optical microscope for the characterization of peptide microarrays has been reported on antireflection substrates [10]. In contrast, the VSI technique is independent of specific substrate coatings suited for glass substrates common for molecular arrays and allows a vertical and lateral resolution down to the nanometer range [11]. A significant enhancement in the spatial resolution of a VSI method was achieved via sub-pixel sample positioning [12] or the use of a dual wavelength white light emitting diode as a light source [13]. The maximal lateral VSI resolution of ~ 20 nm was theoretically predicted [14]. Some studies report the use of VSI to characterize surface roughness of membranes [15] and engineering surfaces [16], as well as topographic and geometric investigations of cells [17]. But, according to our knowledge, the VSI method has not yet been explored for label-free detection of peptide-antibody interactions. 

To estimate the feasibility of the VSI technique for this application, we focus on the measurement of antibody layer thickness accumulated in the high-density array format. For this purpose, the protein layer thickness resulting from the incubation of serum reactive peptide spots with different dilutions of serum and secondary antibody were compared with the fluorescent signal intensities. To evaluate the accuracy of VSI measurements, peptide spots are additionally scanned via an atomic force microscope, and layer profiles of both VSI and AFM measurements are compared.

## 2. Materials and Methods

### 2.1. Array Assembling: Peptide Array, Antibodies and Incubation of Peptide Arrays

Peptide arrays were produced by spotting pre-synthesized peptides containing a C-terminal cysteine onto 3D-Maleimide glass surfaces (PolyAn, Berlin, Germany) (Figure 1 and Figure 2). This surface serves as a model surface for the peptide arrays synthesis. Glycine-serine-glycine-serine was synthesized as a spacer between the C-terminal cysteine and the actual peptide sequence. A serum reactive peptide (MVPEFSGSFPMRGSGSC) [18] and the synthetic peptide containing the hemagglutinin (HA) fragment of an influenza virus (YPYDVPDYAGGSGSC) [19] (Peps4LS, Heidelberg, Germany) were dissolved into phosphate buffered saline (PBS) at pH 7.4 (Sigma-Aldrich, St. Louis, MO, USA) to concentrations of 0.4 mM, 0.2 mM, and 0.1 mM. Tris(2-carboxyethyl)phosphonium chloride (TCEP) was added to 50 μL of a peptide solution. Spotting was conducted with a NanoPlotter 2.1 (GeSiM, Radeberg, Germany) using the Pico Tip J A070-402 p00738A. For each spot, two drops of approximately 400 pl peptide solution were deposited. Peptides were covalently bound by a thiol-maleimide click reaction [20]. After spotting, the slides were dried for 2 h and subsequently blocked in PBS containing 0.4% 2-Mercaptoethanol (Merck, Darmstadt, Germany). The washing was performed at 70 rpm on an orbital shaker (Orbital Shaker DOS-20S, Elmi Ltd., Riga, Lettland) for 3 min with PBS, 3 min deionized water, 5 min acetonitrile (VWR Chemicals, Missouri, TX, USA) containing 0.1% Trifluoroacetic acid (99%) (Honeywell Chemicals, Mexico City, Mexico), 5 min DMF (99.8%) (VWR Chemicals, Radnor, PA, USA) containing 0.5% DIPEA (Merck, Darmstadt, Germany), 3 min × 5 min DMF and 2 min × 3 min Methanol (Merck, Darmstadt, Germany). Slides were dried in an argon gas stream and stored at 4 °C until further usage.

The donation of serum by a human individual was approved by the state chamber of physicians of Baden-Wuerttemberg (reference number: F-2011-044) and informed consent was obtained. To visualize the serum antibodies bound to the serum reactive peptide, a secondary antibody (Goat anti-human IgG (FC γ) AF647, Jackson Immunoresearch, Bothell, WA, USA) was used at a concentration of 13.3 nM. Monoclonal anti-HA antibody, (provided by Dr. Gerd Moldenhauer from DKFZ, Heidelberg, Germany), diluted to 1 µg/µL was conjugated to the fluorophores DL550 NHS esters (DyLight 550 NHS ester, Thermo Fisher Scientific, Waltham, MA, USA) according to the instructions.

Peptide arrays were incubated according to the protocol published by Weber et al. [18] using PBS-T (PBS containing 0.05% Tween 20 (Sigma Aldrich, St. LouisCity, MO, USA)) for the dilution of antibodies. The incubation of primary antibodies was conducted overnight at 4 °C. Unbound antibodies were removed by washing with PBS-T, and the secondary antibody was incubated for 30 min followed by washing. All incubations were conducted under agitation using an orbital shaker with a rotation speed of 140 rpm. The arrays were dried in an argon gas stream, and a fluorescence scan, as well as AFM and VSI measurements, were performed.

### 2.2. Array Fluorescence Scanning

Fluorescence scanning was performed on the Innoscan 1100 AL (Innopsys, Carbonne, France) at 647 nm, gain 3, laser power low, and a resolution of 1 μm/pixel. The resulting 16-bit grayscale images were analyzed using Mapix (Innopsys, Carbonne, France). The median gray scale value of all pixels in a spot was subtracted by the median gray scale value of at least 5 background control areas (spot size of 80–90 µm each), located in the vicinity of the spot of interest.

### 2.3. Vertical Scanning Interferometry (VSI)

Spot topology was imaged by the Contour GT-K by Bruker using Vertical Scanning Interferometry (VSI) in VXI mode for higher z-resolution at ambient conditions.

A scheme of the setup can be seen in Figure 3. VSI uses the interference of two coherent beams of light to determine differences in their optical path lengths. For this purpose, a beam of light is emitted by a light source and separated by a beam splitter, located in the Mirau objective, into a sample and a reference beam. The sample beam passes the splitter and is reflected off the sample surface and directed back to the objective, whereas the reference beam is reflected off the splitter. The beams are directed back together and form an interference pattern of dark and light fringes, which are sampled by a CCD camera. The obtained interferograms are analyzed by software to calculate the surface height pixel by pixel, and topographic 3D models are generated. With 1.7 mm by 1.3 mm dimensions, the setup offers a big field of view in relation to a subnanometer z-resolution. Therefore, a large area scan of an array of 11 × 29 spots is conducted and spot properties are evaluated for quality control. The green light source was selected, the back scan was set to 3 microns, and the number of measurements to average was set to 3 at auto resolution. To reduce noise and improve image quality, the device was operated on a vibration-dampening table. For the 5× objective, the field of view was 1.75 mm by 1.3 mm. To image an array with the area of 3 mm by 1.75 mm, 3 measurements were automatically stitched together, applying an overlap of 23%. Measurements were flattened in Vision64, applying an operational F-filter using the ‘tilt only’ option. Additionally, dust particles were masked via histogram. Median spot heights were obtained by the ‘multiple region’ option, where the leveling was set to auto, and the threshold for height and number of contiguous pixels was set depending on the analysis conducted.

### 2.4. Atomic Force Microscopy (AFM)

All AFM measurements were performed under ambient conditions with a Bruker Dimension Icon AFM. The measurements were conducted in ScanAsyst mode using ScanAsyst-Air cantilevers (Bruker, Billerica, MA, USA) with a resonance frequency of about 150 kHz and a scan speed of 0.15 Hz. The offline image flattening and analysis were performed with the NanoScope Analysis software (Version 8.4, provided via Software Downloads—Service of Bruker, Billerica, MA, USA).

### 2.5. Comparison of AFM and VSI Measurements Force Microscopy (AFM)

For the comparison of spot topologies measured with AFM and VSI, the respective measurements were imported into Gwyddion 2.49 [22]. Minimal z-values were set to zero and a third-degree polynomial background correction was conducted. Data sets were exported as text files and further processed in Matlab 14a. For alignment, the position of a prominent feature in the middle of the spot was used as the zero point. Measurements were set up to be congruent to each other, so no rotational correction was needed in post-processing. This was accomplished by reference markers located some millimeters next to the structures of interest. Considering the respective resolution, line profiles were extracted, intersecting the spots vertically and horizontally.

### 2.6. Determining Antibody Profiles for VSI Measurements

For each level of serum dilution, 9 spots were measured using VSI as described above. The medium spot heights were determined using the multiple region option in Vision64, which identifies all areas matching the threshold criteria in height and amount of contiguous pixels. Thresholds for each dilution level were set to maintain a constant number of pixels per spot.

## 3. Results and Discussion

### 3.1. Fast Scanning Large Field of View Using VSI

A peptide array consisting of 16 subarrays of 11 × 29 peptide spots each was produced, incubated with serum and antibodies and imaged by VSI. Peptides were deposited onto the functional surface by spotting HA peptide and serum peptides at a concentration of 400 μM onto a 3D-Maleimide surface. The serum reactive peptide pattern has the form of the KIT symbol, while other spots, supplementing the rectangular subarray, were HA peptides. The array was incubated with serum (diluted 1:50) and anti-HA antibody DL550 (13.3 fM) overnight and then with fluorescently-labeled anti-human secondary antibody (AF647) at 13.3 nM for 30 min. The concentration of anti-HA antibody added was chosen to be two magnitudes lower than the concentration of secondary antibody to see if the anti-HA antibody was still detectable using VSI. A scheme of peptide deposition and antibody incubation is depicted in Figure 1. The whole slide was scanned by VSI to determine the coordinates of each pattern relative to two corners of the 3D-Maleimide slide and to image the individual patterns by stitching 3 measurements. For perfectly plane and smooth surfaces, areas being multiples of the field of view (1.7–1.3 mm) can be scanned by stitching individual measurements together without readjusting focus. In Figure 4, one of the 16 patterns is shown (see also a corresponding fluorescent image in Appendix A). It was possible to resolve spot heights in a z-range of 7 nm, while the lateral size of the measurement is in the millimeter scale. Small imperfections of the functionalized surface are visible, which would remain undetected using fluorescence images. Even the anti-HA antibody-HA peptide spots are vaguely perceptible in between the plainly visible serum spots in the 3D visualization. The height profile drawn across the array in Figure 4C shows the ratio in height of HA spots to incubated SRP spots, which is about ~1–7. A more detailed evaluation of SPR spot properties is shown in Figure 4D. Applying a threshold of 1 nm and a minimum of 1000 contiguous pixels, all 125 SRP spots are captured. The mean spot height is distributed between 2 and 3.5 nm corresponding to a spot area of 1800–4200 pixels, whereas the spot diameter ranges from 70–100 μm. This rather broad distribution results to some extent from the zero-level applied for the calculation, which was averaged for all non-spot areas.

This experiment, demonstrating the possibility of both the detection of antibody layers in spots in a high-density array format and distinguishing between different antibody binders, reveals a principle question about the physical meaning of the thickness observed via VSI. The antibodies are known to be Y-shaped proteins with the dimensions 14.5 nm × 8.5 nm × 4.0 nm [23]. The observed thickness of the SPR spots in a range of 3 nm is significantly smaller than the thickness of 8 nm in the case of assembling the primary and secondary antibodies flat on each other on the surface. The reason why the measured thickness of antibody layers does not correspond to the dimensions of an IgG might be, for one, due to the ability of the antibody to migrate to some extent into the polymer layer onto which the peptides are immobilized (also peptides migrate into the polymer during immobilization). Apparently, the concentration of antibodies should have a significant influence on the thickness of the antibody layer. To answer this question, the thickness of the accumulated antibody layers was measured at different serum dilutions. 

### 3.2. Qualitative Comparison Between VSI and Fluorescence-Based Detection

The detection of peptide-bound antibody is usually realized via fluorescence since it is very sensitive. The antibody is, therefore, conjugated to a dye acting as a fluorescent label. At higher concentrations of labeled antibody, the fluorescence intensity does not correlate any more with the amount of adsorbed antibody due to a self-quenching effect. Such effects are based on the energy transfer between fluorescent dyes that occurs at distances in the range of <10 nm [10,24,25]. In this section, the relation between the thickness of the accumulating antibody and the fluorescent signals obtained via secondary antibodies in arrays format was studied.

Serum reactive peptide was spotted in 3 dilutions (400 μM, 200 μM, and 100 μM) and incubated with 7 concentrations of serum ranging from 1:800–1:12.5, resulting in 21 areas of 9 spots each (see Figure 2). Human serum was serially diluted from 1:12.5–1:800 in PBS-T (PBS containing 0.05% *v*/*v* Tween 20 (Sigma-Aldrich, St. LouisCity, MO, USA)) and incubated overnight at 4 °C. The slide was therefore divided into separated subarrays to avoid cross-contamination by using a press-on sealing. The secondary antibody was incubated for 30 min. Finally, a fluorescently-labeled secondary antibody was incubated at 13.3 nM. The array was scanned using VSI and fluorescence imaging and obtained spot heights and spot fluorescence intensities for each field were averaged for 9 spots respectively (Figure 5). As expected, peptide spots spotted at 100 μM resulted in lower protein layer thickness than higher peptide concentrations. At a dilution of 1:100 for the 400 μM and 200 μM spots, and 1:50 for the 100 μM spots, no further increase in protein layer thickness can be observed for higher serum concentrations. At the same time, for dilutions lower than 1:100, the fluorescence intensity decreases drastically.

The dependence of the protein layer thickness on the serum dilution can be described by formula (1): (1)$$\Delta h=\Delta h_{sat}\frac{\lambda /2}{\lambda /2+\delta \left(k\right)}$$
Whereby Δ*h* is the thickness and Δ*h_sat_* is the saturation thickness of the accumulated antibody layer, *δ*(*k*) the distance between two macromolecules which is dependent of the serum dilution *k*, and *λ*/2 is the optical resolution limit corresponding to the averaged wavelength *λ* used for the VSI. The physical meaning of this formula consists in the averaging of the protein nanolayer thickness if the accumulated molecules are separated by a relatively large distance. For example, if the distance between two antibodies is *λ*/2, the average thickness of the accumulated layer will be Δ*h_sat_*/2. If the distance between two antibodies is >>*λ*, the VSI will deal with separated molecules and no accumulation layer will be detected: Δ*h* → 0. As the fluorescent excitation/emission and the VSI wavelength are in the visible range of the order of 500 nm and *δ*(*k*) ~ 10 nm in the region of the dye quenching, the ration *δ*(*k*)/*λ* << 1 and, accordingly, the Δ*h* = Δ*h_sat_*. This explains why the saturation region of the antibody thickness curve and the region of the fluorescent dye quenching are considered in the experiment. 

The experimental curves for VSI and fluorescence-based detection exhibit a linear dependence of the measured signals at small *k*. This fact could be supported using Formula (1) and the Michaelis–Menten equation [26] for the fluorescent signal *I* at large dilutions *kN*_0_ << *K_D_* (in our case *k* is in the range of 1:800 to 1:400):(2)$$I=I_{sat}\frac{kN_{0}}{K_{D}}$$
Here, *I_sat_* is the saturation fluorescent signal, *K_D_* the dissociation constant of the peptide bound antibody within the spot, and *N*_0_ is the concentration of the antibody in the serum. Taking into account that *δ* is inversely proportional to the surface density of the accumulated molecules und correspondingly inversely proportional to *I*, the dependence of *δ*(*k*) has the form: *δ* ~ 1/*I* ~ 1/*k*. Thus, Δ*h* ~ *k* in the case of a strong dilution *λ*/2 << *δ* is observed in the experiment. 

Thus, the VSI can be used for measurement of the protein accumulation thickness in a wide range of protein concentrations. In the saturation region, the hight of the VSI profile corresponds to the physical thickness of the protein layer and is independent of the density of the molecules absorbed. In contrast, a linear dependence of layer thickness on the dilution takes place at large dilutions, which can be resolved by VSI. In this case, the label-free VSI measurements can be used for studying protein-protein kinetics in a high-density array format.

### 3.3. Spot Topology is Consistent for Measurements with AFM and VSI

If the VSI values correspond to the thickness of antibody accumulation layer, this thickness should be detectable with alternative methods, such as atomic force microscopy (AFM), which is an established method for surface characterization. For this purpose, a single spot was selected from the experiment in Section 3.2 and measured by AFM with a lateral resolution of 76.1 nm/pixel and VSI with a lateral resolution of 74.6 nm/pixel (Figure 6A). Since the exact coordinates of the spot were determined in previous VSI scans, finding the position for the AFM measurement was easy, and the measurement of the whole area took approximately 90 min. For the VSI measurement, an area of 96.06 μm by 95.91 μm was scanned at 50-fold magnification by stitching two times, resulting in 1288 by 1286 pixels. Both measurements were flattened and z-offset was adjusted as described in Section 2.5. 

Both the AFM and VSI patterns look like concentric circles because the spot was formed via double spotting. As the spotting occurs by contact with the surface, the material volume to be transferred and the spreading of the droplets depend on the surface hydrophilicity. While the first spot was transferred onto the 3D-Maleimide surface, the second spot contacted the material, predominately peptides, transferred by the first spotting. The enhanced concentration of the optical density at the edges of the spots observed in the experiment is known as the “coffee ring” effect [27].

To compare the absolute thickness profiles, 3 vertical and 3 horizontal profiles were extracted from the respective measurements (Figure 6C). In general, the AFM profiles match the VSI profiles along to the selected axes. However, the VSI scan appears to be minimally higher compared to the AFM scan. This was also observed for other spots (see Appendix A). The AFM profiles are noticeably more scattered than the VSI in blue, and for the horizontal profiles, the waviness due to flattening is also visible. This is confirmed by comparing the shape of the respective histogram (see Figure 6B), which is slightly slimmer for VSI, and noticeably jagged with a broader tailing for the AFM. A superimposed frequency can be seen in the AFM based histogram as well. As a result, both histograms cannot be matched exactly. 

## 4. Conclusions

Using VSI, thin protein layer profiles were studied on a large field of view by stitching several scans while maintaining a vertical resolution of 0.1 nm. VSI was successfully applied to measure the accumulation of the fluorescently-labeled secondary antibody in a dilution series of serum. For low antibody concentrations, at which the distance between the absorbed antibodies is comparable with the visible wavelength range, the VSI profiles demonstrate linear dependency on the antibody concentration. This regime may be used for label-free detection of the peptide-protein kinetics in a high-density array format. For large antibody concentrations at which the distance between antibodies is much smaller than the size of optical resolution, the thickness of the antibody film is independent of the further rise of antibody surface concentration. The measurement of the spot topologies via AFM has proved that the thickness obtained via VSI at saturation conditions corresponds to the real physical thickness of the protein layers accumulated in the array format in the spots. 

## Figures and Tables

**Figure 1 high-throughput-08-00007-f001:**
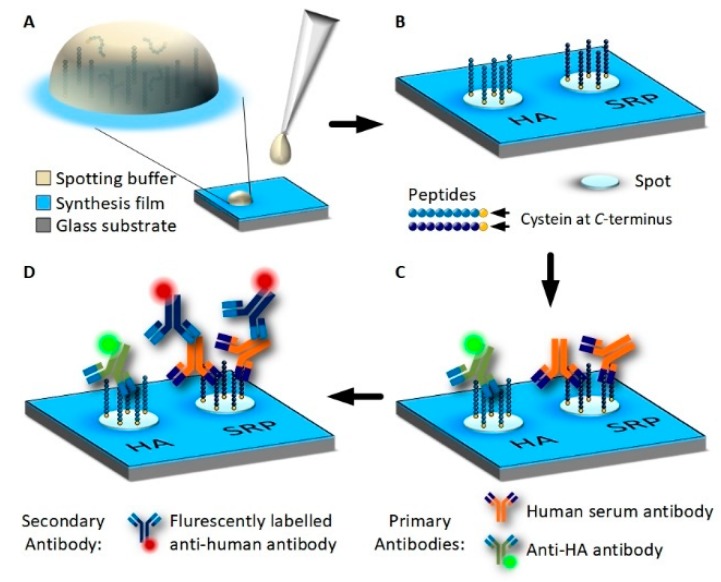
Immobilization of HA-tag and serum reactive peptide (SRP) on 3D Maleimide surfaces using spotting and incubation with antibodies. (**A**): Pre-synthesized peptides are dissolved in PBS and are deposited onto the functionalized surface as drops using a spotting robot. (**B**): Peptides contain a C-terminal cysteine and bind covalently to the synthesis film by a thiol-maleimide click reaction. (**C**): Peptide array incubated with human blood serum and at a low concentration fluorescently labeled anti-HA antibody overnight. Antibodies from the sample bind to the peptides. (**D**): Unbound serum and HA-antibodies are removed by washing and the peptide array is incubated with fluorescently-labeled, anti-human secondary antibody for fluorescence detection.

**Figure 2 high-throughput-08-00007-f002:**
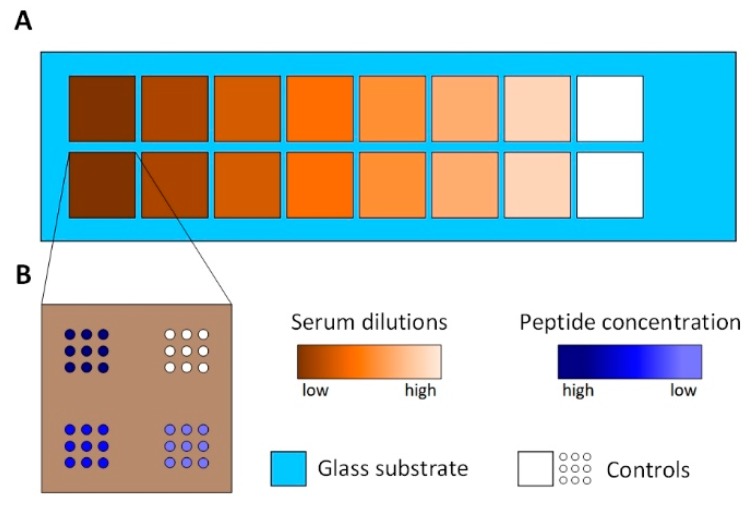
The Peptide array setup. (**A**): The area of the peptide array is divided into 16 subarrays using press-on sealing. Seven different serum dilutions are incubated on two subarrays each. White colored subarrays represent controls incubated with PBS-T. (**B**): The spotting schemes of each subarray. Three different peptide concentrations are spotted with nine spots each. White colored spots represent controls, spotted using PBS only.

**Figure 3 high-throughput-08-00007-f003:**
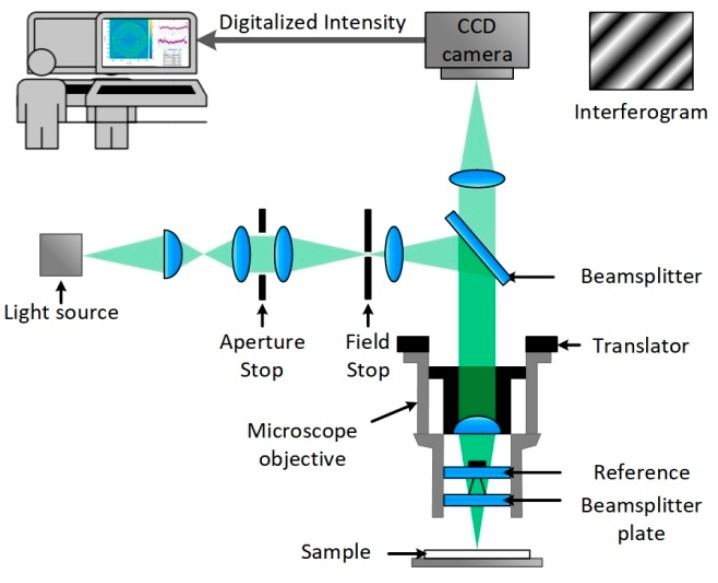
A sketch of a vertical scanning interferometer with Mirau objective and a CCD camera connected to a computer where the obtained interferograms are analyzed by software and topographic 3D models are calculated [21].

**Figure 4 high-throughput-08-00007-f004:**
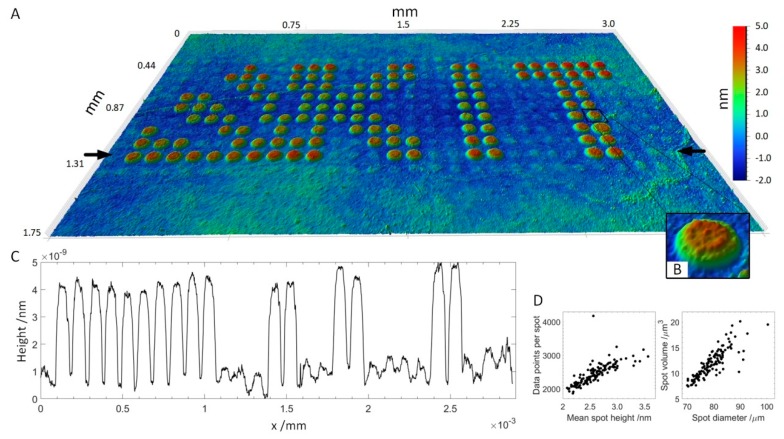
Peptide array incubated with human blood serum and secondary antibody. (**A**): 3D visualization of the array topography measured by VSI. Field of view: 3.0 × 1.75 mm with 1.4 μm lateral resolution. (**B**): Magnification of the spot of interest (SOI). (**C**): Height profile across the array in between the black arrows in A. (**D**): Properties of SRP spots. **Left**: The number of data points per spot as a function of mean spot height. **Right**: Spot volume as a function of spot diameter.

**Figure 5 high-throughput-08-00007-f005:**
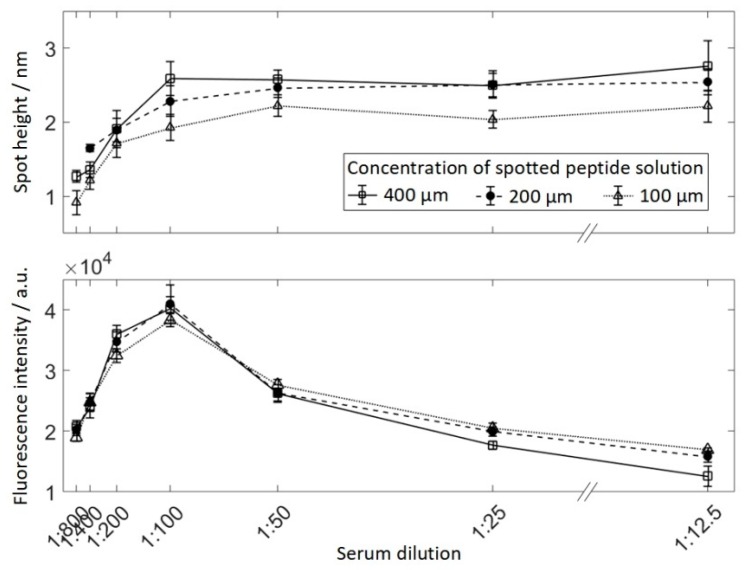
The median fluorescence intensity (**top**) and the median height of spots incubated with serum and secondary antibody for different dilutions of serum. Peptides were spotted in three different concentrations: 400 μM, 200 μM, and 100 μM.

**Figure 6 high-throughput-08-00007-f006:**
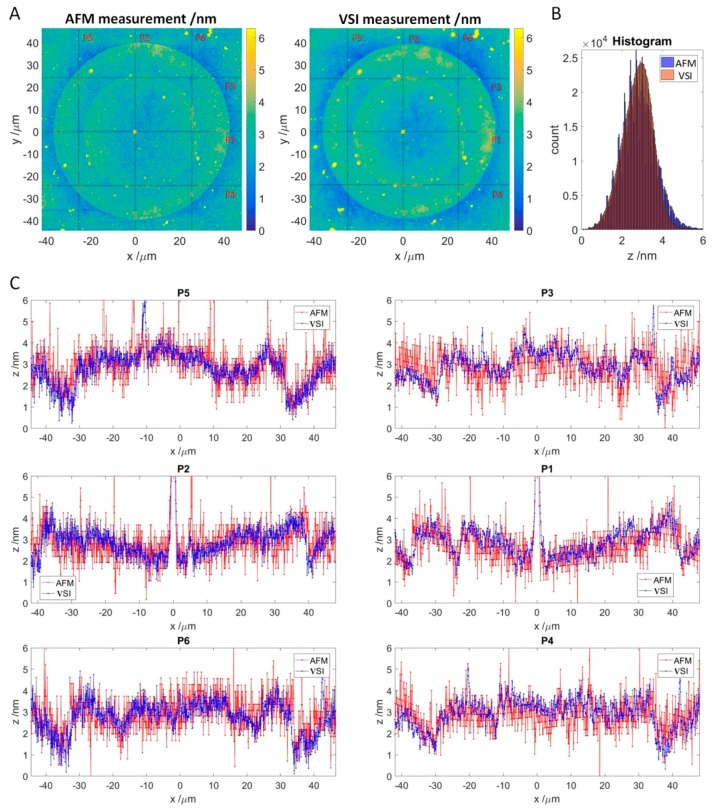
Comparison of the VSI measurement to the AFM measurement. (**A**): Topology of SOI measured by AFM (**left**) and VSI (**right**) with adjusted z-offset. Red lines indicate measured line profiles depicted in C. (**B**): Stacked histograms of AFM and VSI measurements of the SOI. (**C**): Line profiles for the AFM and VSI measurement. AFM profiles are in red, and VSI profiles are in blue. **Vertical**: P5, P2, and P6. **Horizontal**: P3, P1, and P4.

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
