# Peer review of "Vertical Scanning Interferometry for Label-Free Detection of Peptide-Antibody Interactions"

_2571-5135, 2019, doi:10.3390/ht8020007_

Reviewer 1 Report

The authors present vertical scanning interferometry (VSI) as a new means to evaluate peptide-antibody interactions on microarrays. They use a set of two model peptides and matching antibodies to create a model microarray that is examined by VSI and for comparison with atomic force microscopy (AFM) as the standard method for the measurement of heights in the nanometer scale. Comparison of the two data sets shows that VSI is sufficiently accurate to evaluate the height of microarray spots after antibody incubation. In evaluating heights it is much faster than AFM and therefore well suited to evaluate microarrays that cover several square micrometers. It may also be suitable as a label free method to evaluate peptide microarrays, e.g. for the identification of linear epitopes or quantification of specific antibodies in serum. The experiment to assess the dynamic range of peptide microarray evaluation by VSI is not well designed and the chapter describing the findings is not very conclusive. This must be improved before the paper can be published. The revision should also improve some chapters in the methods section and compare the findings to other recent works using VSI. Minor changes - especially with respect to language and layout - are listed in the attached document.

Major issues:

1) The description of the incubation experiment is not clear: Was the peptide microarray simultaneously incubated with serum and anti-HA-tag antibody? The caption of figure 1 and the description of the incubation protocol are partially complementary partially contradicting. At one point the authors mention that the surface was subdivided for incubation, but it is not clear in which way and for which purpose. A schematic of the array layout would be helpful.

2) It is also not clear whether the array shown in figure 2 was incubated with serum and secondary antibody alone or also with anti-HA-tag antibody. The text suggests the later. However, if this was the case, the single antibody layer deposited on the HA peptide spots would be hardly discernible from the background while the layer formed after incubation with serum and anti-human antibody is seven times as high. This is not intuitive and the authors must comment on this observation. The expected high of an antibody-layer should be discussed in this context.

3) The experiment comparing “dynamic range” between VSI and fluorescence intensity scan is not well designed and the description of the results is not conclusive. “Dynamic range” means the difference between lowest and highest detectable concentration. As the antibody concentration in the respective serum sample is unknown, the authors should at least state the range between the corresponding minimal and maximal dilution. This issue would be avoided if the authors had used an antibody in defined concentration, e.g. the fluorescently labeled anti-HA-tag antibody or an unlabeled anti-HA-tag antibody in combination with a fluorescently labeled secondary antibody. Repetition of the experiment with well-defined analyte concentrations is strongly recommended!

In the description of the results the authors should not only discuss the dynamic range but also the linear range in which a concentration (or dilution) could be unambiguously derived from the measured height or fluorescence intensity. They should also comment on the effect (or lack of effect) of concentration of spotted peptide solution. For a quantitative comparison of the two methods the authors should also make use of the Z’-factor that is commonly used to evaluate the suitability of an assay for high-throughput (see Zhang, J. Biomol Screen. 1999). The heading of the respective chapter should be changed to point out that this is about a comparison of VSI and fluorescence intensity scanning.

4) In the conclusions the authors write about the impact of flattening algorithms. This does not become clear in the results section and should be described in more detail there.

Minor issues:

1) In the introduction a reference for the first introduction of VSI should be included in line 44.

2) The authors claim that the studies described in references 11-13 are in the micrometer range. Reference 12 clearly describes results in the submicron range (50 -500 nm).

3) The height resolution should also be compare to more recent publications (e.g. Arvidson et al Microsc. Microanaly 2014; Little & Kane Opt. Express. 2013; Chong et al. Appl. Opt. 2013).

4) With respect to Figure 4 the authors should explain, why the spots look like concentric circles. It should also be commented that the edges of the microarray spot can be hardly recognized in the height profiles as the intensity beyond the edges of the spot assumes values almost as high as inside the spot.

5) The application of VSI for label-free microarray-based assays should be outlined in more detail in the conclusions section, especially its advantages and mode of evaluation.

6) The figure captions of the supporting material don’t match the text cited in the manuscript.

7) Figure S4 has a poor resolution and needs to be replaced.

8) Minor corrections of language and layout are listed in the attached document.

Author Response

Please see the response in the attached file.

the authors.

Reviewer 2 Report

A. Palermo et al. have reported interesting concept and application of vertical scanning interferometry ((VSI) to provide label-free measurement of protein layer topographies on peptide array spots. Their concept of applying VSI to peptide array is certainly interesting, and their logic that label-based peptide array assay has difficulties handling the noise and artifact from the labeled molecule or probe itself.

However, I could not see the clearly that this VSI technique is worth replacing such conventional fluorescent labeling technique.

There are several aspects that makes this work not overwhelming the conventional technique.

First, for the journal "high-throughput", this article do not really provide the data of their high-throughput performance of this technique. The time effectiveness, or cost effectiveness, or handling effectiveness. 

Second, they say that fluorescent labeling makes auto-fading which inhibit the higher concentration assay or detailed assay. But also with their data in Fig. 5, their measurement data seems "equilibrated". Therefore, their measuring resolution does not promise for higher protein accumulation. Moreover, if it the "flattened surface of accumulation=accumulation has stopped from VSI data" is reflecting the true fact, they should reveal that with other method that "no more accumulation is occuring". How can they deny whether the interferometry work/do not work well with multi-layered or aggregated protein condition? How can they say that there are really no increase of accumulated layer thickness?

Third, I think they should provide more understandable description whether they are measuring the "spot height" or "spot height + accumulated protein layer height". With Fig.3 it is very misleading. And more over the thickness of protein accumulation data is clear. I cannot simply follow the difference whether their "spot height" is meaning what.

I do think that if they are assay the "accumulation profiles" of proteins on various proteins or peptide spots, that should be very effective and informative. However, if this is designed to understand  "protein-protein interaction", it is not correct. They are just trying to measure the "thickness". 

Therefore, I think, total concept and objective should be re-written, if the with to further consider in this journal.

I feel this manuscript is nearly Reject, but if authors eager to re-write their concepts and add their "thickness" data and discussion with major revise, I think this work will show good impact.

Author Response

Please see the response in the attached file.

the authors.

Round  2

Reviewer 2 Report

Authors have revised the manuscript with a careful description for the interpretation of their assay concept. I feel their manuscript now has good impact and clarity. I am happy to agree that this manuscript is now has satisfied the level of acceptance.